# AI-Assisted Exploratory Causal Modeling of Cumulative Advantage in Small-$N$ Domains

**ChatGPT-5**
OpenAI

**Zach Huang**
The Overlake School
zhuang@overlake.org

## Abstract

This paper develops an AI-assisted computational framework for exploratory causal modeling of cumulative advantage in small-$N$, spatially heterogeneous domains, demonstrated through a case study of junior golf. The analysis is methodologically challenging due to sparse state-level data, high collinearity among predictors, and the need to approximate unobservable factors. We address these challenges with a **dual-method framework** that combines forward-selection regression with leave-one-out cross-validation (LOOCV) for predictive modeling, and Directed Acyclic Graph (DAG)-guided structural modeling to explore assumption-dependent associations and simulate counterfactual scenarios.

Using data on 16,000+ junior golfers across all U.S. states, we find that population and participation serve as strong baseline predictors of elite performance; PGA Tour event presence, a proxy for elite training access, shows an independent and sizable association; financial strength is predictive only for Top 50 girls; and climate shows little direct association once other factors are accounted for. Exploratory simulations—such as adding a PGA Tour event or increasing participation—suggest potential gains in elite-player production. Our framework demonstrates how AI-assisted exploratory causal modeling can generate transparent, assumption-guided insights that generalize beyond sport to other small-$N$ scientific domains.

## 1 Introduction

The book *Outliers* by Malcolm Gladwell [14] popularized the concept of the Relative Age Effect (RAE)—a phenomenon where individuals born earlier in a selection year enjoy cumulative advantages, particularly in youth sports, due to greater physical and psychological maturity. While often discussed in the context of team sports such as hockey and soccer, where early physical development is heavily rewarded, RAE is more broadly a lens into **cumulative advantage**—how initial benefits compound over time to create significant long-term disparities.

Golf presents an intriguing counterpoint. Prior studies [5, 8, 23, 33] suggest that RAE is largely absent at the elite levels of golf, such as the PGA and LPGA tours. The individual nature of the sport, its emphasis on skill over physicality, and flexible age-group structures may dilute age-based disparities [13]. Even in collegiate golf, where some evidence of RAE emerges under a calendar-year cutoff, the effect is modulated by athletes' ability to take a gap year—effectively neutralizing age disadvantages [7, 30]. This raises a critical question: **If not relative age, are there other hidden cumulative advantages in golf?**

One such potential advantage is geography—specifically, the climate and competitive environment where a golfer grows up. Studies and expert commentary [4, 9, 24, 34] point to a strong concentration of top elite golfers in warm-weather states such as Florida, Texas, California, and Arizona. These regions not only offer year-round access to training but also host a high density of tournaments and professional-level infrastructure. Yet, this climate-based explanation has inconsistencies. Some states with similar weather and infrastructure do not produce proportionally elite talent, and unlike early-year birth, **climate is a movable condition**—families can relocate.

To rigorously test the impact of geography and climate, we turn to **junior golfers**, a population uniquely positioned for analysis. Unlike adults, junior players typically cannot move or make career decisions independently. If climate or geography acts as a cumulative advantage, its effects should be most pronounced here. Nonetheless, this approach brings added complexity: junior golfers also lack control over financial resources and coaching access, the effect of which needs to be carefully isolated. Furthermore, the scarcity of public data on junior golf makes rigorous empirical analysis more difficult than with professional or NCAA players.

Analyzing such structural advantages also presents a methodological challenge: the data are sparse (only 51 state-level observations), highly collinear (e.g., population, participation, and income), and many relevant factors are unobservable. Standard regression approaches risk overfitting or obscuring theoretically meaningful effects. To address this, we develop a **dual-method computational framework**: a forward-selection pipeline with leave-one-out cross-validation (LOOCV) for predictive robustness, paired with structural modeling guided by a directed acyclic graph (DAG) to clarify assumptions and explore how geography, participation, and financial strength may shape outcomes. While our methods cannot by themselves establish causality, they provide a transparent framework for interpreting adjusted associations and simulating plausible counterfactual scenarios. Cumulative advantage manifests across domains, but is especially challenging to study in small-$N$, structurally heterogeneous contexts. Junior golf provides a compelling testbed for developing methods that can generalize to other small-$N$, structurally heterogeneous domains.

We summarize our contributions below:

- **Framework**: We introduce an AI-assisted dual-method computational approach that combines LOOCV-based regression for predictive robustness with DAG-guided structural modeling for transparent assumption testing.
- **Findings**: Participant base and PGA Tour presence explain over 85% of performance variance. PGA presence consistently shows an independent association, while climate and income associations are largely mediated.
- **Exploratory Insights**: Under our DAG assumptions, PGA event access appears linked to enhanced outcomes beyond participation. Purchasing power is predictive for Top 50 girls but shows a stronger conditional association for boys.
- **Implications**: Assumption-dependent simulations suggest hosting PGA events yields the largest performance gains, participation programs offer moderate improvements, and targeted aid may support players on the cusp of elite status.
- **AI in Science**: The work demonstrates how AI systems can support hypothesis generation, model selection, and exploratory causal analysis, enabling reproducible and interpretable scientific workflows.

## 2 Related Work

**Econometrics and AI-Inspired Modeling**. Our methodological approach draws inspiration from econometric and causal-inference traditions, particularly the use of **DAGs** to clarify assumptions and guide adjusted association estimates. Rather than treating DAGs as sources of definitive causal identification, we use them to support **exploratory causal modeling under explicit assumptions**, in line with recent AI- and ML-inspired approaches to structured inference [2, 22, 20]. Pearl [26] provides the foundational framework for DAG-based reasoning, while Arlot & Celisse[1] and Hastie, Tibshirani & Friedman [17] motivate our use of cross-validation and forward-selection regression in small-sample, multicollinear settings.

**Cumulative Advantage in Sports and Economics**. This work also connects to the sports economics and sport science literatures on cumulative advantage. In economics, Szymanski [32], Groot [16], and Fort & Maxcy [12] analyze how structural features and competitive balance shape outcomes, while Humphreys & Ruseski [18, 19] model sport participation as an economic decision influenced by income and opportunity costs. In sport science, studies of climate [28, 29], coaching access [3, 21], and the relative age effect [5, 11] demonstrate systemic influences on athlete development. However, these works often stop short of structural or assumption-guided modeling. Our framework builds on these strands by combining predictive accuracy with assumption-aware structural analysis, using junior golf as a case study.

# 3    Methodology

We develop an integrated framework that maps observable proxies for structural factors (Section 3.1), builds predictive models using forward-selection regression (Section 3.2), and estimates adjusted associations under explicit assumptions (Section 3.3) based on a domain-informed DAG (Sectioni 3.4).

AI assistance was integrated throughout this pipeline: generating candidate hypotheses, suggesting DAG structures, and drafting the initial implementations of regression and cross-validation routines. Human oversight refined these outputs, ensuring alignment with domain expertise in junior golf and methodological standards. This collaboration situates our framework as an example of AI-assisted exploratory causal modeling in practice.

## 3.1    Potential Factors

We consider five key factors that may create cumulative advantages for junior golfers across states:

- **Climate** influences playable days and year-round access to golf.
- **Coaching and Training Access**, at both entry-level (e.g., number of courses) and elite-level (e.g., PGA event infrastructure), shapes development pipelines.
- **Financial Strength** affects a junior player's ability to afford travel, training, and tournament exposure.
- **Population** serves as a baseline predictor—larger states have more potential golfers.
- **Participation Base** captures actual engagement, bridging population and performance.

To operationalize these, we map each factor to one or more observable variables:

- Climate—Solar irradiance (average annual $W/m^2$ [6])
- Coaching (entry-level)—Number of golf courses in the state [25, 34].
- Coaching (elite-level)—Number of PGA Tour events hosted in 2025 [4], as a proxy for elite coaching facilities due to the **industrial clustering effect** [15, 27]. For completeness, we also consider the number of LPGA Tour events.
- Financial Strength—We use both (1) state Median Household Income (MHI) and (2) average hometown MHI of junior players to normalized for price levels [3, 10]. We use purchasing power ratio (average hometown MHI / state MHI) to normalize for interstate economic disparity.
- Population and Participation—We use both (1) total state population and (2) number of players who have ever entered a ranked junior golf tournament.

## 3.2    Predictive Modeling Framework

We adopt a forward-selection regression pipeline to identify the most predictive subset of features for elite performance (Top 50, 100, 200 players per state). This serves two goals: (1) **Model explanatory power**: Which factors predict success, and how well? And (2) **Handle multicollinearity**: Control redundant or collinear predictors.

The steps are:

1. Start with the single most predictive variable (by Pearson $r$).
2. Iteratively add the next-best variable if:
    - It significantly improves **leave-one-out cross-validation** (LOOCV) $R^2$ [1, 17, 31], and
    - Its regression coefficient is significant at $p < 0.05$.
3. Stop when no further variables meet both criteria.

This allows a compact, interpretable model suitable for small-$N$, high-collinearity domains. We now present the **General Forward-Selection Regression**:

$$\text{TopPlayers}_s = \beta_0 + \beta_1 \cdot \text{PGAEvents}_s + \beta_2 \cdot \text{Participants}_s + \beta_3 \cdot \text{PurchasingPower}_s + \epsilon$$

Where $s$ indexes U.S. states. Note that not all terms are included in every model; selection is determined by the forward-selection criterion described above.

### 3.3 Sequential Regression for Adjusted Associations

We estimate the conditional contribution of each predictor using multiple regression models that include both the variable of interest and one or more covariates. This approach allows us to isolate the association of a variable $X$ (e.g., PGA Tour events or purchasing power) on the outcome $Y$ (e.g., Top 50 players), while holding a known confounder $Z$ (e.g., participant count) constant.

We implement this using standard linear regression:

$$Y = \beta_0 + \beta_X X + \beta_Z Z + \epsilon$$

where the coefficient $\beta_X$ captures the marginal association of $X$ on $Y$, controlling for $Z$. Statistical significance is assessed via $p$-values, and the magnitude of $\beta_X$ is used to support interpretation of explanatory and policy-relevant power.

This method provides an interpretable estimate of adjusted relationships and aligns with our forward-selection regression framework. While it does not, on its own, establish causal claims, it serves as the empirical engine for exploratory interpretation of conditional associations under a DAG.

### 3.4 DAG-Guided Structural Modeling and Back-Door Adjustment

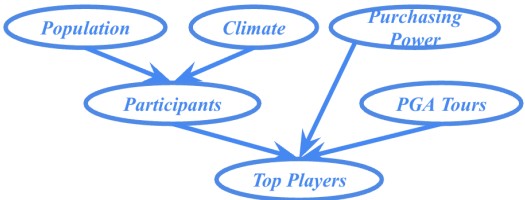

Figure 1: DAG of assumed relationship armong variables

To move from adjusted association toward structured interpretation, we formalize our assumptions using a DAG (Figure 1). The DAG encodes our domain-informed view of how population, participation, income, and access to elite tournaments may interact. This representation does not guarantee causal identification but instead clarifies the assumptions required for exploratory causal modeling. Using back-door adjustment and sequential regression, we examine conditional relationships under these assumptions and simulate potential structural changes. The DAG is constructed from domain expertise and temporal logic:

- **Population** and **climate** influence participation levels, but not vice versa.
- **PGA Tour event presence** is assumed to be an exogenous institutional decision that precedes and facilitates elite development, not a function of junior player counts.
- **Purchasing power** is treated as a fixed background factor derived from family socioeconomic status.

Given this structure, we identify valid **adjustment sets** using the back-door criterion [26], which specifies that all non-causal (confounding) paths from a treatment variable $X$ to an outcome $Y$ must be blocked by conditioning on a sufficient set of covariates $Z$.

We focus on two key relationships:

- Estimating the association of **PGA Tour presence** on top player counts, adjusting for **participants**

- Estimating the association of **purchasing power**, also adjusting for **participants**

These lead to the following adjusted association model:

$$\text{TopPlayers}_s = \alpha_0 + \alpha_1 \cdot X_s + \alpha_2 \cdot \text{Participants}_s + \nu_s$$

Where $X \in \{\text{PGAEvents}, \text{PurchasingPower}\}$ depending on the variable of interest, and $s$ indexes states.

Under the assumed DAG, the coefficient $\alpha_1$ can be interpreted as an adjusted association (under assumptions) of $X$ on elite performance, enabling us to explore the implications of structural interventions under these assumptions (e.g., hosting more PGA events or increasing financial support).

# 4 Data

We describe the sources and collection methods for our dataset in Section 4.1, including both publicly available and manually curated variables. In Section 4.2, we detail how we transform these raw data into structured, state-level inputs.

## 4.1 Raw Data

We collected data from a combination of public sources and manually curated datasets. State-level indicators—such as total population, average solar irradiance, number of golf courses, number of PGA and LPGA Tour events hosted in 2025, and state median household income—were obtained from publicly accessible sources. Specifically, population figures and median household incomes are published by the U.S. Census Bureau, while solar irradiance data is available through national climatological datasets and solar resource atlases.

City-level median household income data, required for assessing players' relative purchasing power, is also publicly available from the U.S. Census Bureau's portal (data.census.gov). Player-level data—specifically national ranking and hometown—was sourced from juniorgolfscoreboard.com, which maintains public profiles for 12,344 boys and 4,550 girls.

## 4.2 Data Preparation

To enable correlation analysis, we transformed the raw player-level data into aggregate state-level metrics. For each U.S. state (including D.C.), we computed:

- Total number of junior players with known hometowns ("participants")
- Number of players ranked in the national Top 50, Top 100, and Top 200

This transformation allows us to analyze how observable state-level variables (climate, income, coaching access, etc.) correlate with both broad participation and elite performance in junior golf.

For the financial strength factor, we computed each state's **average junior player purchasing power**. This was derived from the set of hometown-level median household incomes associated with players from each state. Let $S$ denote a state with $K$ players, and let $I_k$ be the median household income of the $k$-th player's hometown, where $k \in [1, K]$. The average hometown income for the state is then $\bar{I}_S = \frac{1}{K} \sum_{k=1}^{K} I_k$.

To remove the association of differing price levels and income baselines across states, we normalize this quantity by dividing it by the state's median household income $MHI_S$. The result is the **relative purchasing power** of junior golfers in that state $PP_S = \frac{\bar{I}_S}{MHI_S}$.

This normalization allows us to compare the financial standing of junior players across states in a way that controls for interstate economic disparities. A higher $PP_S$ implies that junior players in that state, on average, come from relatively wealthier cities compared to the state's economic baseline. If a state has no players, we use a trivial $PP_S = 1$.

Table 1: LOOCV Analysis for Participants

| Category | Selected Variables | LOOCV $R^2$ |
|---|---|---|
| Boy Participants | Population | 0.843 |
| Girl Participants | Population, PGA | 0.904 |

## 5 Results and Findings

We evaluate the predictive and explanatory power of state-level factors on junior golf participation and elite performance for boys and girls separately. We structure the results in three parts: predictive performance (Section 5.1 and 5.2), controlled associations (Section 5.3), and counterfactual simulation (Section 5.4).

### 5.1 Predictive Performance: Participants

Our forward selection regression pipeline achieves strong predictive accuracy despite the limited number of U.S. states ($N = 51$). As shown in Table 1, both models achieve a LOOCV $R^2 \geq 0.8$ in predicting the number of participants.

The best-performing models consistently include **population**, indicating that participant count is largely driven by state population. For boys, population is the only variable selected into the model. Notably, **solar irradiance**—our proxy of climate—does not enter the model for either group. Similarly, the number of **LPGA Tour events** is not selected, even for the girls' model, suggesting these variables have limited explanatory power for participation at the state level.

### 5.2 Predictive Performance: Top $N$ Players

For Top $N$ players, all but one model achieve a LOOCV $R^2 \geq 0.5$, with the sole exception being borderline ($R^2 = 0.493$) (Table 2). This indicates strong out-of-sample explanatory power, especially considering the inherent noise and complexity of observational social data.

The best-performing models consistently include **number of PGA Tour events** and **participant count**, validating their role as primary structural predictors of elite performance.

Despite having relative strong correlation with Top 50 rankings (Table 6 and 7), **Purchasing Power** does not enter the final model for boys. In fact, it only appears in the model for **Girls' Top 50**, suggesting that its predictive power is limited and context-specific.

For boys, top-ranked players are consistently explained by a combination of **PGA Tour event presence** and **participant count**. In contrast, the predictors for girls exhibit more nuanced variation: the **number of golf courses** emerges as a stronger signal. This divergence points to subtle differences in the competitive structure between boys' and girls' junior golf—potentially reflecting differences in access, developmental pathways, or institutional support.

Table 2: LOOCV Analysis for Top $N$ Players

| Category | Selected Variables | LOOCV $R^2$ |
|---|---|---|
| Boys' Top 50 | PGA, Participants | 0.500 |
| Boys' Top 100 | PGA, Participants | 0.765 |
| Boys' Top 200 | PGA, Participants | 0.887 |
| Girls' Top 50 | PGA, Purchasing Power | 0.493 |
| Girls' Top 100 | Participants, Golf Courses | 0.594 |
| Girls' Top 200 | Participants, Golf Courses, PGA | 0.751 |

Table 3: Control Analysis. Each cell list the controlled association $\beta_X$ (and its $p$-value) of $X$ controlling on Participants, where $X \in \{\mathrm{PGA, Solar, PurchasingPower}\}$. Numbers crossed out means not statistically significant.

| Category | PGA | Solar | Purchasing Power |
|---|---|---|---|
| Boys' Top 50 | $\beta_{\mathrm{PGA}} = 0.824, p = 7.04e-3$ | ~~$p = 0.802$~~ | $\beta_{\mathrm{PP}} = 4.28, p = 4.14e-4$ |
| Boys' Top 100 | $\beta_{\mathrm{PGA}} = 1.68, p = 2.43e-5$ | ~~$p = 0.268$~~ | ~~$p = 0.258$~~ |
| Boys' Top 200 | $\beta_{\mathrm{PGA}} = 3.54, p = 4.19e-8$ | ~~$p = 0.321$~~ | ~~$p = 0.590$~~ |
| Girls' Top 50 | $\beta_{\mathrm{PGA}} = 1.15, p = 4.78e-3$ | ~~$p = 0.623$~~ | $\beta_{\mathrm{PP}} = 1.19, p = 0.0273$ |
| Girls' Top 100 | $\beta_{\mathrm{PGA}} = 1.85, p = 7.794-3$ | ~~$p = 0.976$~~ | ~~$p = 0.563$~~ |
| Girls' Top 200 | $\beta_{\mathrm{PGA}} = 3.37, p = 1.09e-3$ | ~~$p = 0.757$~~ | ~~$p = 0.134$~~ |

## 5.3 Controlled Associations

To better understand inter-variable dependencies, we analyze **sequential regression residuals** (Table 3):

- For both genders, the coefficient **# PGA Tour events** remains statistically significant when controlling for participation, with adjusted association $\beta_{\mathrm{PGA}} \approx 0.824 - 3.54, p < 0.01$, suggesting that elite infrastructure contributes predictive value beyond simple scale effects.

- **Climate (solar irradiance)** loses statistical significance after controlling for participation, indicating that its association is likely mediated through participation, not direct performance gains.

- **Purchasing power** (average hometown MHI normalized by state MHI) is statistically significant for **Top 50 players**, but the strength of the association is notably larger for boys ($\beta_{\mathrm{PP}} = 4.28, p = 4.14e-4$) than girls ($\beta_{\mathrm{PP}} = 1.19, p = 0.0273$). This suggests a **gendered disparity** in how financial advantage translates to elite outcomes.

These results support a multi-pathway model of cumulative advantage, with **infrastructure and participation** as dominant channels, and **financial capacity** playing a narrower, top-end role—especially for boys.

Interestingly, **Purchasing Power** did not enter the predictive model selected by forward-selection and LOOCV, suggesting that—within the full pool of predictors—it does not offer substantial additional explanatory power for cross-validated prediction. However, when analyzed independently using the DAG-guided model controlling for participant count, it exhibits a **statistically significant and strong association**, particularly for boys' Top 50 rankings.

This divergence is not contradictory: predictive modeling selects variables that *improve generalization in the presence of multicollinearity*, whereas exploratory causal modeling helps highlight the **conditional association** of a theoretically motivated variable. In small-$N$ settings with correlated predictors, a meaningful structural factor can be **masked** in a predictive pipeline. This reinforces the value of using both approaches in tandem.

Non-significant predictors, such as LPGA Tour events and state-level median income, are detailed in Appendix Table 8 and 9.

## 5.4 Counterfactual Simulation

To illustrate implications of these adjusted associations, we simulate counterfactual scenarios for both policymakers and parents.

**For Policy Makers**: To illustrate policy implications, we simulate counterfactual scenarios using the fitted regression models (parameters in Table 3):

- **Adding a PGA Tour event** in an average state is projected to increase Top 50 player count by approximately **84.05% − 117.3%**, controlling for participation. It has a comparable impact on Top 200 player count by approximately **85.94% − 90.27%**.

- **Boosting participation by 20%** (e.g., through grassroots programs or school golf teams) predicts a $15.38\% - 17.64\%$ increase in Top 200 players for boys and $14.35\% - 18.16\%$ for girls.
- **Increasing purchasing power by 10%** (e.g., through targeted financial aid) is projected to increase Top 50 representation by **43.66% for boys** and **12.14% for girls**.

These simulations highlight that **hosting elite tournaments is associated with the largest gains**, with notably higher numbers of top-ranked juniors. Broadening participation through grassroots access produces more modest but consistent gains across both genders. Financial support, while less impactful on average, has a strong association for boys at the Top 50 level and a smaller but still positive association for girls. Together, these results suggest that structural investments deliver the highest aggregate returns, while targeted financial aid may help unlock elite potential in specific groups.

**For Parents**: While the counterfactual simulations are framed from the perspective of policymakers (e.g., adding PGA events), the same regression coefficients can be interpreted at the individual level. For example, a parent relocating their junior golfer to a state with one additional PGA Tour event, or one with 20% more junior participants, would experience similar predicted gains in top-player odds—*assuming all else is equal*. Likewise, increasing household income by 10% improves relative purchasing power, and can be interpreted similarly to receiving targeted financial aid. Nonetheless, these interpretations rely on the assumption that the player's *competitive position remains consistent* after relocation—an assumption that may not hold in all cases.

# 6 Discussion

Our findings have several practical implications for parents, coaches, and policymakers.

**First**, parents should not be discouraged if their child develops an interest in golf while living in a state without a strong golf tradition. Given that **population** emerged as the leading predictor of participation for both boys and girls, it is natural that children in more populous states are more likely to be drawn into the sport. Moreover, the **number of participants** in a state consistently ranked among the top two predictors of elite performance. This suggests that all players have almost equal chances of reaching the top.

**Second**, access to **elite training institutions and facilities** holds the highest predictive power for producing top-ranked juniors. States hosting PGA Tour events tend to have clusters of championship-level courses and associated high-caliber coaching resources, which create cumulative advantages for player development. While relocating to such states may offer tangible advantages, it is not the only pathway to accessing elite training and coaching opportunities. Advances in technology—such as launch monitor systems (e.g., TrackMan) and high-definition swing analysis—combined with structured **remote coaching** can help mitigate geographic disadvantages.

**Third**, financial strength demonstrated **only modest predictive power**, and only for players in the **Top 50**. For policymakers, tournament organizers, and grant programs such as the AJGA ACE Grant, this finding offers guidance: targeted financial assistance could yield the highest return on investment when focused on juniors with the potential to break into the top tier. Such targeted support could help talented players overcome the last barriers to national prominence, particularly in contexts where financial constraints might otherwise limit competitive opportunities.

# 7 Conclusion

In this study, we modeled how structural factors relate to junior golf performance across U.S. states. Our exploratory simulations suggest that, under these assumptions, hosting PGA Tour events is associated with the largest gains in elite player production, while participation programs deliver smaller but consistent improvements. Targeted financial support shows a sizeable conditional association for boys at the Top 50 level and a smaller positive association for girls. These results should be interpreted as assumption-dependent rather than definitive, but they illustrate how structured modeling can generate actionable insights and guide further research on equity and resource allocation in youth sports.

## 8 AI Agent Setup

All analyses and interactions in this study were conducted using **ChatGPT-5** equipped with the **Deep Research** capability. No customized orchestration pipelines or external tool integrations were employed. While Deep Research was available, it was selectively invoked depending on the complexity and openness of each query. To maintain conceptual coherence within the primary research thread, we instantiated separate chat sessions for auxiliary prompts (e.g., *"Explain what Foo is"* or *"How to apply Foo to my data"*). This deliberate branching strategy was intended to minimize contextual interference and preserve a consistent line of reasoning across iterative exchanges. Although a formal A/B evaluation has not been performed, our qualitative experience suggests that such compartmentalization enhances session stability and epistemic continuity.

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

Table 4: The Pearson correlation coefficients between each candidate factor and the number of boy participants in each state, along with associated $p$-values for the correlation, and $p$-value for the forward selection. Numbers crossed out means not statistically significant.

| # Boy Participant | Population | # PGA | # Courses | # LPGA | Solar | State Income |
|---|---|---|---|---|---|---|
| Correlation | 0.930 | 0.864 | 0.817 | 0.535 | 0.299 | $-0.0521$ |
| $p$-value corr | 0 | 0 | 0 | $5.27e-4$ | 0.0332 | ~~0.716~~ |
| $p$-value fwd sel | 0 | 0.0135 | 0.0389 | ~~0.186~~ | $7.67e-3$ | |

Table 5: The Pearson correlation coefficients between each candidate factor and the number of girl participants in each state, along with associated $p$-values for the correlation, and $p$-value for the forward selection. Numbers crossed out means not statistically significant.

| # Girl Participant | Population | # PGA | # Courses | # LPGA | Solar | State Income |
|---|---|---|---|---|---|---|
| Correlation | 0.936 | 0.926 | 0.760 | 0.537 | 0.292 | $-0.0103$ |
| $p$-value corr | 0 | 0 | $1.01e-10$ | $4.92e-5$ | 0.0375 | ~~0.943~~ |
| $p$-value fwd sel | 0 | $1.50e-7$ | ~~0.455~~ | ~~0.0810~~ | 0.0160 | |

# A   Participation and Nonsignificant Factors

Table 4 and 5 list the results for boys and girls participants, respectively. Table 6, 7, 8, and 9 list the factors for boys and girls top players, respectively.

# B   Responsible AI Statement

Our research was conducted with a commitment to ethical principles in AI and data science, as outlined in the NeurIPS Code of Ethics. Key measures include:

- **Privacy Protection & Consent**: All data sources used are publicly available and aggregated at the state or city level. No personally identifiable information (PII) was collected or analyzed. Player data includes only hometown locations, not names or identities.

- **Bias Awareness & Fairness**: We acknowledge potential biases in demographic and participation variables—such as financial inequality or regional disparities affecting access to junior golf resources. We explicitly interpret our findings as descriptive of systemic patterns rather than prescriptive assessments of individual ability, and include a section discussing structural inequities and gender differences uncovered in the analysis.

- **Transparency & Reproducibility**: The modeling pipeline, including forward selection, LOOCV evaluation, and causal adjustment via DAGs, is fully documented in the paper.

Table 6: The Pearson correlation coefficients between each candidate factor and the number of top 50, 100, 200 boy players in each state, along with associated $p$-values for the correlation, and $p$-value for the forward selection. Numbers crossed out means not statistically significant. To save space, number of LPGA tour, solar irradiance, and state median household income are omitted and moved to Table 8

| Boy Top $N$ | Metrics | # PGA | # Participants | Population | # Course | Pur Pw |
|---|---|---|---|---|---|---|
| | Correlation | 0.806 | 0.802 | 0.726 | 0.627 | 0.755 |
| 50 | $p$-value corr | 0 | 0 | $1.66e-9$ | $8.54e-7$ | $1.60e-10$ |
| | $p$-value fwd sel | 0 | $5.78e-3$ | ~~0.823~~ | ~~0.414~~ | $8.01e-3$ |
| | Correlation | 0.895 | 0.882 | 0.818 | 0.667 | 0.364 |
| 100 | $p$-value corr | 0 | 0 | 0 | $8.92e-8$ | $8.55e-3$ |
| | $p$-value fwd sel | 0 | $1.79e-4$ | 0.0332 | ~~0.375~~ | ~~0.0621~~ |
| | Correlation | 0.929 | 0.910 | 0.856 | 0.656 | 0.321 |
| 200 | $p$-value corr | 0 | 0 | 0 | $1.70e-7$ | 0.0217 |
| | $p$-value fwd sel | 0 | $5.17e-6$ | 0.0391 | ~~0.0542~~ | ~~0.111~~ |

Table 7: The Pearson correlation coefficients between each candidate factor and the number of top 50, 100, 200 girl players in each state, along with associated $p$-values for the correlation, and $p$-value for the forward selection. Numbers crossed out means not statistically significant. To save space, number of LPGA tour, solar irradiance, and state median household income are omitted and moved to Table 9

| Girl Top $N$ | Metrics | # Participants | # PGA | Population | # Course | Pur Pw |
|---|---|---|---|---|---|---|
| 50 | Correlation | 0.850 | 0.864 | 0.746 | 0.482 | 0.430 |
| | $p$-value corr | 0 | 0 | $3.28 - 10$ | $3.47e - 4$ | $1.65e - 3$ |
| | $p$-value fwd sel | 0.0359 | 0 | 0.0232 | 0.0103 | $7.62e - 3$ |
| 100 | Correlation | 0.885 | 0.885 | 0.787 | 0.513 | 0.296 |
| | $p$-value corr | 0 | 0 | 0 | $1.19e - 4$ | 0.0350 |
| | $p$-value fwd sel | 0 | $3.89e - 3$ | 0.0310 | $1.01e - 3$ | ~~0.395~~ |
| 200 | Correlation | 0.922 | 0.919 | 0.842 | 0.581 | 0.292 |
| | $p$-value corr | 0 | 0 | 0 | $7.89e - 6$ | 0.0376 |
| | $p$-value fwd sel | 0 | $5.44e - 4$ | ~~0.121~~ | $9.71e - 4$ | ~~0.339~~ |

Table 8: The Pearson correlation coefficients between each candidate factor and the number of top 50, 100, 200 boy players in each state, along with associated $p$-values for the correlation, and $p$-value for the forward selection. Numbers crossed out means not statistically significant. Only show number of LPGA tour, solar irradiance, and state median household income which are omitted in Table 6

| Boy Top $N$ | Metrics | # LPGA | Solar | State Income |
|---|---|---|---|---|
| 50 | Correlation | 0.542 | 0.260 | $-0.142$ |
| | $p$-value corr | $4.02e - 5$ | ~~0.0651~~ | ~~0.319~~ |
| | $p$-value fwd sel | ~~0.505~~ | | |
| 100 | Correlation | 0.533 | 0.335 | $-0.111$ |
| | $p$-value corr | $5.61e - 5$ | 0.0161 | ~~0.437~~ |
| | $p$-value fwd sel | ~~0.198~~ | ~~0.138~~ | |
| 200 | Correlation | 0.479 | 0.329 | $-0.125$ |
| | $p$-value corr | $3.82e - 4$ | 0.0186 | ~~0.382~~ |
| | $p$-value fwd sel | ~~0.202~~ | ~~0.115~~ | |

Table 9: The Pearson correlation coefficients between each candidate factor and the number of top 50, 100, 200 girl players in each state, along with associated $p$-values for the correlation, and $p$-value for the forward selection. Numbers crossed out means not statistically significant. Only show number of LPGA tour, solar irradiance, and state median household income which are omitted in Table 7

| Girl Top $N$ | Metrics | # LPGA | Solar | State Income |
|---|---|---|---|---|
| 50 | Correlation | 0.431 | 0.208 | 0.102 |
| | $p$-value corr | $1.61e - 3$ | ~~0.143~~ | ~~0.478~~ |
| | $p$-value fwd sel | ~~0.320~~ | | |
| 100 | Correlation | 0.424 | 0.257 | 0.0615 |
| | $p$-value corr | $1.91e - 3$ | ~~0.0691~~ | ~~0.668~~ |
| | $p$-value fwd sel | ~~0.465~~ | | |
| 200 | Correlation | 0.447 | 0.253 | 0.0432 |
| | $p$-value corr | $1.02e - 3$ | ~~0.0737~~ | ~~0.763~~ |
| | $p$-value fwd sel | ~~0.356~~ | | |

We based our implementation exclusively on open-source libraries (pandas, NumPy, SciPy, scikit-learn, statsmodels). We will publicly release the code along with detailed instructions, and make the manually curated dataset (including hometown median household income estimates) available post-review to support reproducing the results.

- **Responsible Data Release**: The curated dataset will exclude any personal identifiers and will be shared under a permissive license. A data documentation template (i.e., a Datasheet for Datasets) will accompany the release to clarify the data's provenance, limitations, and intended use.

- **Societal Impact Considerations**: We discuss both positive impacts—such as guiding equitable resource allocation and training support—and potential negatives, including the risk of reinforcing geographic segregation or socioeconomic stratification. We emphasize that relocation or intervention strategies should account for broader social contexts and should be implemented with care.

By adhering to these standards, we aim to ensure that our research is ethically rigorous, transparent, and socially responsible.

## C  Reproducibility Statement

All methods and models used in this paper are fully specified in the main text, including the use of forward-selection regression, leave-one-out cross-validation (LOOCV), and sequential regression with back-door adjustment. We implemented the entire modeling pipeline using standard, open-source Python libraries: pandas, numpy, scipy, scikit-learn, and statsmodels. These tools were selected for their reliability and widespread adoption in reproducible data science.

We will publicly release all code upon publication, along with detailed instructions for reproducing our experiments, tables, and figures. In addition, we will share the manually curated dataset—which includes state-level variables and city-level median household income estimates for junior golfers' hometowns. These materials will be accompanied by metadata and documentation to ensure that other researchers can verify and build on our findings. Together, these efforts are intended to support full transparency and enable faithful replication of all results.

## D  Limitations

This study is subject to several limitations, stemming from both data constraints and methodological assumptions.

**Data Accessibility and Granularity**: While the analysis leverages a sizable dataset of junior golfers, several important variables—such as coaching quality, frequency of play, and family support—are unobservable and had to be approximated using proxies like PGA Tour presence or income at the city level. In addition, player-level income was estimated based on hometown median household income, which may not accurately reflect individual circumstances.

**Manual Data Curation**: Due to platform limitations (e.g., authentication gating on juniorgolfscoreboard.com and anti-bot restrictions on data.census.gov), key components of the dataset were manually curated. This includes more than 400 entries of city-level income data, which increases the risk of transcription errors and limits reproducibility.

**Causal Assumptions**: The causal inferences in Section 3.4 rely on assumed DAG structure and conditional independence assumptions that are not empirically validated. While we apply standard back-door adjustments using well-motivated covariates, the analysis does not rule out unmeasured confounding or feedback loops. The causal estimates should therefore be interpreted as exploratory rather than definitive.

**Limited Sample Size**: With only 51 observations at the state level, statistical power is inherently limited. While we mitigate overfitting using LOOCV and forward selection, the small sample size restricts our ability to detect weaker effects and generalize findings to finer-grained geographies (e.g., city or school district level).

**Model Simplicity and Scope**: We adopt linear regression models for interpretability and stability, but this excludes non-linear relationships or interaction effects that may exist between factors (e.g.,

income × population). Future work could explore richer models such as causal forests or Bayesian networks, if more granular data becomes available.

These limitations highlight the trade-offs involved in studying complex social phenomena with incomplete data, and underscore the importance of combining statistical evidence with domain expertise.

## Agents4Science AI Involvement Checklist

1. **Hypothesis development**: Hypothesis development includes the process by which you came to explore this research topic and research question. This can involve the background research performed by either researchers or by AI. This can also involve whether the idea was proposed by researchers or by AI.

   Answer: [C]

   Explanation: The initial research question—exploring whether cumulative advantages exist in junior golf beyond the known Relative Age Effect—was proposed by the human author. In response, the AI generated most of the candidate hypotheses, including climate, financial access, coaching infrastructure, and tournament exposure. The human author reviewed these and contributed two key additions: (1) introducing **state-level population** as a baseline or null hypothesis, and (2) suggesting **PGA Tour presence** as a proxy for access to elite-level training infrastructure. While the AI carried out the majority of the mapping from abstract factors to observable variables, the human author also proposed using **city-level median household income** to represent a junior player's relative financial standing. Overall, the hypothesis development process was a collaborative effort, with AI driving hypothesis generation and variable formulation, and the human author contributing domain-specific insight and critical validation.

2. **Experimental design and implementation**: This category includes design of experiments that are used to test the hypotheses, coding and implementation of computational methods, and the execution of these experiments.

   Answer: [C]

   Explanation: The human author designed the overall analysis strategy and executed the initial experiments, including all correlation analyses and variable curation. More than 7 hours were spent manually collecting and structuring key datasets—particularly city-level household income and player ranking data—that could not be accessed programmatically. The human author also implemented the full pipeline for basic statistical comparisons. AI was used to assist in the design of advanced statistical components, including the forward-selection regression framework, leave-one-out cross-validation (LOOCV), and causal modeling via DAG-based adjustment. These methods were selected and carried out using the AI's knowledge of advanced statistical modeling, under the human author's supervision. The initial implementation was generated by AI, while the human author contributed substantial effort in debugging and integrating the results into the broader workflow.

3. **Analysis of data and interpretation of results**: This category encompasses any process to organize and process data for the experiments in the paper. It also includes interpretations of the results of the study.

   Answer: [B]

   Explanation: AI led the execution of advanced statistical methodologies, including forward-selection regression, leave-one-out cross-validation (LOOCV), and DAG-based causal modeling. However, the insights generated through these methods diverged from the initial correlation analysis conducted by the human author. This prompted careful scrutiny, validation, and interpretation. The human author played a central role in critically evaluating the results, cross-referencing them with the raw data, and contextualizing all conclusions through domain-specific knowledge acquired over years of firsthand experience in junior golf competition.

4. **Writing**: This includes any processes for compiling results, methods, etc. into the final paper form. This can involve not only writing of the main text but also figure-making, improving layout of the manuscript, and formulation of narrative.

   Answer: [C]

   Explanation: The writing process was a collaborative effort between the human author and AI. **Section 1 (Introduction)** was primarily written by AI, following a storyline outlined by the human author. AI also drafted the summary of contributions based on the overall structure and key insights defined by the human. **Section 2 (Related Work)** was outlined by the human author, while AI conducted the literature review and drafted the content. In **Section 3 (Methodology)**, AI designed the advanced statistical methodology and authored the majority of the corresponding sections, including those on predictive modeling and

causal inference. **Section 4 (Data)** was mostly written by the human author, who also performed extensive manual data collection. Although AI was tested for data curation tasks, it was unable to scale to larger queries (e.g., hundreds of player hometowns). **Section 5 (Results)** The structure of the section was initially drafted by AI, with the human author responsible for statistical summarization and interpretation. AI then incorporated the data and completed the writing based on the finalized analyses. **Section 6 (Discussion)** was again primarily written by AI, following a storyline outlined by the human author.

5. **Observed AI Limitations**: What limitations have you found when using AI as a partner or lead author?

    Description: A few key limitations were observed during the collaboration with AI. **First**, AI was unable to automate large-scale data collection tasks—such as retrieving median household income for several hundred player hometowns—due to rate limits, authentication requirements, and lack of robust scraping support. These tasks had to be completed manually by the human author. **Second**, while the initial version of the code was generated by AI, the human author invested significant time in integrating and debugging it to ensure correctness and compatibility with the overall workflow. **Third**, in multi-turn prompting workflows (especially with ChatGPT's DeepResearch mode), the AI occasionally over-indexed on the most recent instruction and lost context from earlier, well-structured outputs. The human author had to repeatedly reiterate prior guidance or copy-paste earlier content to ensure continuity and consistency across revisions. At times, the AI produced responses that were entirely unrelated to the prompt, requiring manual redirection or correction by the human author. **Fourth**, AI-generated citations are often fabricated or incorrect, and therefore require manual verification using reliable academic search engines or databases.

