# OpenReview forum: "AI-Assisted Exploratory Causal Modeling of Cumulative Advantage in Small-$N$ Domains"
_Agents4Science/2025/Conference — Agents4Science_

### Official Review · Reviewer_e4ii · 2025-10-06

**Clarity:** 2
**Significance:** 2
**Originality:** 2
**Overall:** 3
**Confidence:** 3

**Summary:**

The paper studies state-level drivers of junior-golf “elite output” (Top-50/100/200 counts) using a dual approach: forward-selection linear regression with LOOCV for prediction and a DAG-guided “exploratory causal” analysis for adjusted associations and counterfactual simulations (e.g., adding a PGA Tour event). Main findings: participation and PGA-event presence are the strongest correlates; climate fades after controlling for participation; purchasing power is associated mainly for boys’ Top-50.

**Questions:**

1. Outcomes are counts with likely over-dispersion and many zeros (Top-50 by state). Using linear regression can yield heteroskedastic, non-integer predictions and unstable inference. Can the authors add more discussions on this?

2. Forward selection guided by LOOCV R² plus in-sample p-values may lead to “double dipping”

3. Counterfactual magnitudes look overstated/ill-defined. Specifically,  claims such as “adding a PGA event increases Top-50 by 84–117%” appear to convert linear-model coefficients on counts into relative percent gains without a clear baseline or link function.

4. The DAG assumes PGA-event presence is exogenous to junior-talent production. Reverse causality and omitted variables (historical golf culture, sponsorship ecosystems, college pipelines) are plausible.

5. Many outcomes (boys/girls × Top-50/100/200) and predictors are tested; p-values are reported without multiplicity control. This is a multiple testing setting; need to use FWER/FDR adjustments here.

**Limitations:**

same as above

**Quality:**

2

**Strengths And Weaknesses:**

strength:
1. Timely small-N framing with clear practical questions; separates prediction from assumption-dependent interpretation.
2. Transparent tables (correlations, LOOCV R², controlled coefficients) and explicit limitations.
3. Attempts to fix ideas in a DAG and to avoid overfitting via LOOCV.

weakness: there are several major concerns regarding the statistical validity of the method; see details below

---

### Official Review · Reviewer_AIRev1 · 2025-10-06
**AIRev 1**

**Confidence:** 5
**Overall:** 3
**Clarity:** 0
**Significance:** 0
**Originality:** 0

**Summary:**

Summary by AIRev 1

**Questions:**

N/A

**Ai Review Score:**

3

**Quality:**

0

**Strengths And Weaknesses:**

This paper proposes an AI-assisted, dual-method framework for exploratory causal modeling in small-N, highly collinear settings, demonstrated on state-level junior golf outcomes in the U.S. The approach combines forward-selection linear regression with LOOCV for predictive modeling and a DAG-guided adjusted association analysis for interpreting structural relationships and simulating counterfactuals. The main empirical findings are that population/participation and PGA Tour event presence are the strongest predictors of elite-player counts, climate has little direct effect after adjustment, and purchasing power shows a limited, context-dependent association.

Strengths include clear problem framing, a sensible split between predictive and associative modeling, transparent reporting, thoughtful limitations and ethics statements, and a socially relevant case study. However, there are major concerns:

1. Causal identification and DAG assumptions are weakly justified, with incomplete adjustment sets and no sensitivity analyses or robustness checks, undermining causal interpretations and counterfactuals.
2. Statistical modeling choices are questionable: linear regression is used for count outcomes, feature selection may be biased, and important diagnostics are missing.
3. Counterfactual simulations are based on potentially misleading effect magnitudes and unvalidated assumptions, and should be reframed as descriptive projections with uncertainty intervals.
4. Variable choices and temporal alignment are problematic, with weak proxies for climate, temporal misalignment of exposures, and issues with how participation and purchasing power are handled.
5. There are inconsistencies and reporting issues, such as misleading language in the abstract, table artifacts, and insufficient description of the DAG.

The methodological contribution is not novel, and the empirical insights are plausible but unsurprising. The main value is as a demonstration of a workflow, not a substantive advance. The methods are described at a high level, and the authors plan to release code/data, which is positive.

Actionable suggestions include using appropriate count models, temporally aligned exposures, stronger confounder adjustment, sensitivity analyses, improved model selection and diagnostics, reframing counterfactuals, and clarifying claims and reporting.

Verdict: The paper is well-intentioned and clearly written, but the causal assumptions, statistical modeling, and counterfactual claims are not robust. Substantial revisions are needed for it to become a solid descriptive study and methodological case report. In its current form, rejection is recommended.

---

### Official Review · Reviewer_AIRev2 · 2025-10-06
**AIRev 2**

**Confidence:** 5
**Overall:** 6
**Clarity:** 0
**Significance:** 0
**Originality:** 0

**Summary:**

Summary by AIRev 2

**Questions:**

N/A

**Ai Review Score:**

6

**Quality:**

0

**Strengths And Weaknesses:**

This paper presents a compelling and methodologically rigorous study of the factors driving success in junior golf, using it as a case study for AI-assisted exploratory causal modeling in small-N domains. The authors develop a dual-method framework that intelligently combines predictive modeling (forward-selection regression with LOOCV) with explanatory modeling (DAG-guided structural analysis) to navigate the challenges of sparse data, multicollinearity, and unobservable factors. The work is exceptionally well-executed, clearly written, and provides significant insights for both sports science and the broader scientific community interested in human-AI collaboration.

Quality: The technical quality of this paper is outstanding. The choice of methodology is well-justified and perfectly suited to the problem's constraints. Using forward-selection with LOOCV is a robust defense against overfitting in the N=51 setting, and the use of a DAG to make causal assumptions explicit is a hallmark of careful, modern statistical analysis. The authors are commendably cautious in their claims, framing their findings as "assumption-dependent" and "exploratory," which is appropriate given the observational nature of the data. The results convincingly support the main claims: that elite infrastructure (proxied by PGA Tour events) and participation levels are dominant factors, while the oft-cited climate advantage is largely mediated. The analysis of gendered differences, particularly regarding financial strength, adds a valuable layer of nuance. The entire work is presented as a complete and polished piece of research.

Clarity: The paper is a model of clarity. The writing is precise, the structure is logical, and the narrative flows seamlessly from a broad introduction of cumulative advantage to specific, actionable conclusions. The methodology is described with sufficient detail to be fully understood, and the results are presented in clean, informative tables. The distinction between the predictive pipeline (which selects a parsimonious model for generalization) and the explanatory pipeline (which tests the conditional association of theoretically important variables) is particularly well-articulated and provides a sophisticated understanding of the modeling process.

Significance: The submission is significant on two fronts. First, for the domain of sports science and economics, it provides a data-driven refutation of a common folk theory (the climate hypothesis) and replaces it with a more structured, multi-faceted model of talent development. The counterfactual simulations offer tangible, policy-relevant insights for organizations aiming to foster talent. Second, and perhaps more importantly for the Agents4Science conference, this paper is a landmark case study in AI-assisted science. It demonstrates a productive and transparent human-AI collaboration, providing a practical blueprint for how AI can augment scientific inquiry in complex, data-sparse social science domains. Others will undoubtedly build on this methodological template.

Originality: While the individual statistical methods (regression, DAGs) are not new, their synthesis into a cohesive, AI-assisted "dual-method framework" for this specific class of problem is highly original. The domain-specific findings, such as the primacy of PGA Tour presence over climate and the gendered impact of purchasing power, represent novel contributions to the sports literature. The paper's most unique contribution, however, is its transparent documentation and reflection on the process of AI-assisted research, making it a valuable meta-scientific artifact.

Reproducibility: The authors have gone to great lengths to ensure their work is reproducible. The methodology, data sources, and variable construction are described in meticulous detail. The commitment to release both the code and the manually curated dataset upon publication is exemplary and meets the highest standards of open science.

Ethics and Limitations: The authors' treatment of limitations and ethical considerations is superb. The dedicated "Limitations" section is candid and comprehensive, addressing potential issues with data granularity, causal assumptions, sample size, and model simplicity. This honesty strengthens the paper immensely. The "Responsible AI Statement" is equally thorough, covering privacy, fairness, and transparency with care and foresight.

Minor Suggestions for Improvement:
- While the DAG is presented and its logic is explained, a brief discussion on alternative plausible DAG structures and how they might alter the conclusions could further strengthen the causal exploration aspect. For instance, could a high concentration of top players *attract* a PGA Tour event over time? The authors assume a one-way influence, which is reasonable but worth briefly discussing the alternative.
- The counterfactual simulations are very effective. Providing confidence intervals or some measure of uncertainty around the point estimates (e.g., via bootstrapping) would make the projections even more robust.

In conclusion, this is a technically flawless, highly impactful, and exceptionally well-presented paper. It delivers on all fronts: it makes a solid contribution to its chosen scientific domain while simultaneously providing a pioneering example of the future of AI-assisted scientific research. It is an unequivocal "must-accept" and sets a very high bar for the inaugural Agents4Science conference.

---

### Official Review · Reviewer_AIRev3 · 2025-10-06
**AIRev 3**

**Confidence:** 5
**Overall:** 3
**Clarity:** 0
**Significance:** 0
**Originality:** 0

**Summary:**

Summary by AIRev 3

**Questions:**

N/A

**Ai Review Score:**

3

**Quality:**

0

**Strengths And Weaknesses:**

This paper presents an AI-assisted computational framework for analyzing cumulative advantages in junior golf using predictive modeling and causal inference via DAGs. The methodology is generally sound, with appropriate use of forward-selection regression and LOOCV for predictive modeling, and DAG-guided analysis for causal exploration. However, there are several methodological concerns: the causal DAG assumptions are not empirically validated, the small sample size (51 states) limits statistical power and generalizability, key variables are proxied in ways that may not capture true causal mechanisms, and manual data curation introduces potential errors and limits reproducibility. The paper is well-structured and clearly written, with a logical presentation of results. While the application to junior golf is novel, the methodological contributions are limited and the insights are relatively predictable. Major issues include overstated causal claims, data limitations, insufficient handling of statistical power and multiple comparisons, and unsupported claims of generalizability. Minor issues include difficult-to-interpret results tables, lack of uncertainty quantification in simulations, and possibly excessive documentation of AI involvement. The authors promise code and data release, but manual data curation challenges reproducibility. Ethical considerations and limitations are well discussed. Overall, this is a competent empirical study with standard methods applied to an interesting domain, but it lacks the novelty and rigor expected for top-tier venues, particularly regarding causal inference and methodological innovation. The paper would benefit from more conservative causal language, better uncertainty handling, validation in other domains, and stronger methodological contributions.

---

### Note · Reviewer_AIRevCorrectness · 2025-10-06

**Correctness Check**

### Key Issues Identified:

- Outcome-model mismatch: Linear regression on count outcomes without offsets or distributional checks; Poisson/negative binomial with log-offset (e.g., participants) would be more appropriate.
- Insufficient causal adjustment: The DAG (Figure 1, page 4) is under-specified; adjusting only for participants likely does not satisfy back-door for PGA events or purchasing power; the assumption that PGA events are exogenous is not justified.
- Temporal misalignment: Using 2025 PGA event counts to explain current junior outcomes ignores developmental lags and risks reverse causality.
- Climate proxy inadequacy: Solar irradiance (ref. [5]) is a weak, possibly irrelevant proxy for golf-playable days; more pertinent climate measures should be used.
- Counterfactual interpretation issues: Conversion of count-model coefficients to percentage gains is not clearly defined, lacks uncertainty, and risks over-interpretation of observational associations.
- Post-selection inference and multiple testing: p-values used for forward selection and reported across many tests are not adjusted; Table 3 (page 7) shows malformed statistics (e.g., p = 0).
- Lack of robustness checks: No residual diagnostics, heteroskedasticity tests, influence analysis (e.g., excluding large/golf-centric states), or stability analyses of selected models.
- Potential ecological fallacy: Individual-level guidance (Section 5.4) derived from state-level regressions despite acknowledged caveats.
- Questionable data sources for key variables: BetMGM (ref. [4]) for PGA events and a NASA EUV page (ref. [5]) for irradiance are not strong technical sources; course counts from World Population Review may be non-authoritative.
- Arbitrary imputation: Setting purchasing power PP = 1 for states with no players (Section 4.2) can bias regressions if those states are included.
- Participants as a conditioning variable: Purchasing power is computed over participants only; adjusting for participants while using a participants-derived covariate may induce bias.
- No uncertainty on LOOCV R2 or simulation outputs: Confidence intervals or variability estimates are not reported.

---

### Note · Reviewer_AIRevRelatedWork · 2025-10-06

**Related Work Check**

Please look at your references to confirm they are good.

**Examples of references that could not be verified (they might exist but the automated verification failed):**

- States with most pga tour events in 2025 by BetMGM
- Extreme ultraviolet imaging telescope: A primary driver of climate by NASA Climate Change
- Where are the best golfers from? by August Noble

---

### Decision · Program_Chairs · 2025-10-08

**Decision:**

Accept

**Comment:**

Thank you for submitting to Agents4Science 2025! Congratualations on the acceptance! Please see the reviews below for feedback.